# Gains or Losses in Forest Productivity under Climate Change? The Uncertainty of $CO_2$ Fertilization and Climate Effects

**Dominik Sperlich [1],\*** , **Daniel Nadal-Sala [2,3], Carlos Gracia [3,4], Jürgen Kreuzwieser [5], Marc Hanewinkel [1] and Rasoul Yousefpour [1]**

[1] Forestry Economics and Forest Planning, Faculty of Environment and Natural Resources, University of Freiburg, 79106 Freiburg, Germany; marc.hanewinkel@ife.uni-freiburg.de (M.H.); rasoul.yousefpour@ife.uni-freiburg.de (R.Y.)

[2] Karlsruhe Institute of Technology, Institute of Meteorology and Climate Research—Atmospheric Environmental Research (IMK-IFU), 82467 Garmisch-Partenkirchen, Germany; daniel.sala@kit.edu

[3] Department d'Ecologia, Facultat de Biologia, Universitat de Barcelona, Diagonal 645, 08028 Barcelona, Spain; cgracia@ub.edu

[4] Ecological and Forestry Applications Research Centre (CREAF), Cerdanyola del Vallès, 08193 Barcelona, Spain

[5] Department of Ecosystem Physiology, University of Freiburg, Georges-Koehler-Allee 53/54, 79110 Freiburg, Germany; juergen.kreuzwieser@ctp.uni-freiburg.de

\* Correspondence: dominik.sperlich@ife.uni-freiburg.de

**Abstract:** Global warming poses great challenges for forest managers regarding adaptation strategies and species choices. More frequent drought events and heat spells are expected to reduce growth and increase mortality. Extended growing seasons, warming and elevated $CO_2$ ($eCO_2$) can also positively affect forest productivity. We studied the growth, productivity and mortality of beech (*Fagus sylvatica* L.) and fir (*Abies alba* Mill.) in the Black Forest (Germany) under three climate change scenarios (representative concentration pathways (RCP): RCP2.6, RCP4.5, RCP8.5) using the detailed biogeochemical forest growth model GOTILWA+. Averaged over the entire simulation period, both species showed productivity losses in RCP2.6 (16–20%) and in RCP4.5 (6%), but productivity gains in RCP8.5 (11–17%). However, all three scenarios had a tipping point (between 2035–2060) when initial gains in net primary productivity (NPP) (6–29%) eventually turned into losses (1–26%). With $eCO_2$ switched off, the losses in NPP were 26–51% in RCP2.6, 36–45% in RCP4.5 and 33–71% in RCP8.5. Improved water-use efficiency dampened drought effects on NPP between 4 and 5%. Tree mortality increased, but without notably affecting forest productivity. Concluding, cultivation of beech and fir may still be possible in the study region, although severe productivity losses can be expected in the coming decades, which will strongly depend on the dampening $CO_2$ fertilization effect.

**Keywords:** GOTILWA+; drought; forest growth simulation; $CO_2$ fertilization; European beech; silver fir

## 1. Introduction

Forest productivity in Europe has generally increased over the past decades [1–4], providing that other factors were not limiting it, such as water availability, growth temperature and/or nitrogen deposition [5,6]. Heat spells combined with extended drought periods are, however, increasing worldwide and have negatively affected productivity [7], increased mortality [8,9] and induced shifts in species' distribution ranges and species composition [10–12]. Temperature increase may also positively



affect forest productivity, for example, due to the positive effect on photosynthesis in mountains or high latitudes, which are energy limited and not water limited [13], and also due to the lengthening of the growing season [14,15]. Besides, the continuous increase in atmospheric carbon dioxide (eCO$_2$) acts as a fertilizer for plant growth—termed the "CO$_2$ fertilization effect" [16] because current CO$_2$ levels are far from being limiting for the photosynthetic carboxylation reactions in the chloroplasts of C3 plants [17].

Considerable uncertainties remain regarding the impacts and interplay of temperature, precipitation and eCO$_2$ on forest growth and productivity—especially over longer periods of time, considering the long life-spans of trees [18]. Other uncertainties stem from the different responses of forest ecosystems and underlying ecophysiological processes themselves (Figure 1).

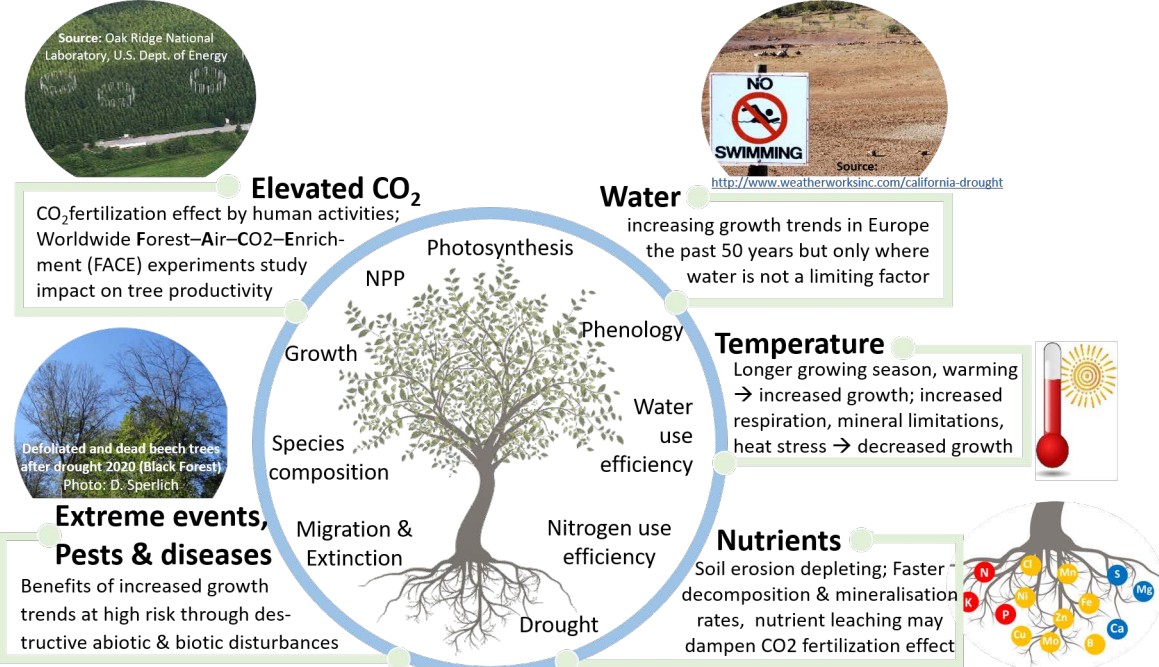

**Figure 1.** Summary of major ecological and ecophysiological factors associated with climate change and their impacts on various biological processes in trees (modified after [19]).

Simulated productivity and biomass carbon pools mainly responded positively to climate change, especially when the effects of increasing CO$_2$ were included [20–23]. The CO$_2$ fertilization effect may be transitory, as shown by free air CO$_2$ enrichment (FACE) experiments, especially in mature stands [22,24,25]. This may be because of photosynthetic downregulation due to acclimation and nutrient limitation ([26,27], but see [28]), changes in carbon allocation patterns [29] and/or increases in ecosystem respiration [24,30]. Nonetheless, eCO$_2$ may still (indirectly) benefit plants by improving their water-use strategy [31,32] and potentially increasing their tolerance to water deficits [33–36], but it most likely will not mitigate the effects of extreme drought events [37,38].

In this study, we focus on silver fir (*Abies alba*) and European beech (*Fagus sylvatica*) (hereafter fir and beech for simplicity) because they are involved in discussions about large-scale transformation and adaptation strategies in Germany to replace Norway spruce (*Picea abies*). The future of these two species is, however, highly uncertain. The drought-susceptibility of beech is a debated question [39–43]. Silver fir generally benefits from a warmer climate, but drought years, especially in combination with secondary agents (bark beetles), can lead to increased mortality [44], as also recently witnessed in the Black Forest [45].

We applied the highly mechanistic process-based model GOTILWA+, which can not only accurately describe the ecophysiology of forests under a variety of environmental conditions (and especially under

drought), but disposes a management module for forestry applications. It has been applied in a wide range of conditions in boreal, temperate, Mediterranean and tropical regions [46–50]. A prerequisite for benefiting from detailed but parameter-heavy models such as GOTILWA+ is a sound parametrization with ecophysiological data and/or Bayesian inference techniques [49]. For example, a high foliar biomass leads to more photosynthesizing tissue—which increases growth. A lower photosynthesizing potential and/or a higher foliar respiration rate can, however, reduce growth and compensate for a higher foliar biomass (calculated from leaf area index—LAI and leaf mass per area—LMA). Despite their importance, they are rarely measured for modelling exercises, and literature values are used instead, which often do not represent the conditions of ecosystems aimed to build up in the simulations.

As a first objective, we aimed to (i) create a set of key ecophysiological variables of silver fir and European beech in field experiments (photosynthetic potential, LMA and LAI, soil and climate) and parametrize GOTILWA+; (ii) to validate the model with regional yield tables to make it applicable for forest management; (iii) to estimate the impacts of climate change on growth, productivity and mortality in a potential risk area for beech and fir under future climate; and (iv) to disentangle the effect of $eCO_2$ from climate effects.

## 2. Material and Methods

### 2.1. The Biogeochemical Forest Growth Model GOTILWA+

GOTILWA+ (Growth Of Trees Is Limited by WAter, http://www.creaf.uab.es/gotilwa/) is a detailed process-based biogeochemical model that simulates tree growth, and the associated carbon and water fluxes, to investigate the effects of tree stand structure, management interventions, soil properties, water stress and climate change—and is also $CO_2$ sensitive (Figure S1 and Note S4 in Supplementary Materials, and see the dynamic scheme at https://prezi.com/to-nd8yjmbaa/gotilwa-a-process-based-forest-growth-model/). GOTILWA+ simulates a monospecific population distributed in classes of diameter at breast height (DBH). All trees in one DBH class produce the same results summarized to the stand level. Hourly ecosystem carbon and water fluxes are estimated using meteorological forcing, which are integrated to daily, monthly and yearly time steps. Drought is simulated by directly coupling the photosynthetic potential through a nonlinear relation to soil water content by using an empirical β coefficient [48,51] (Note S4 in Supplementary Materials). The mortality submodule uses the carbon balance approach in which the balance of demand (growth, maintenance) and supply (photosynthesis, stores) of carbohydrates from the mobile C pool decides over tree mortality [52,53]. Demand depends on growth, maintenance and repair, and supply depends on mobile C-stores and refill from photosynthetic carbon assimilation (Note S4 in Supplementary Materials). The link of the growth process with drought is often not implemented in such a mechanistic manner in other land-surface models compared to GOTILWA+ [51,54].

A management module describes the thinning regime, mode and intensity that can be applied in different diameter classes. The produced diameter classes allow the categorization and monetarization of the standing and harvested wood volume in a separate step with species-specific price and cost tables. GOTILWA+ has been validated against different data sets from boreal, temperate and Mediterranean regions for evergreen broadleaved and conifers and deciduous species [46–49,52,53].

### 2.2. Parametrization with Ecophysiological Field Data

We initialized the modelled stand from an experimental forest near Freiamt located in Southwest Germany at 440 m a.s.l. (48°08.863' North 7°54.331' East) dominated by beech (61%) and silver fir (32%) (others 7%). The tree density was 618 trees/ha. The stand age was 55 years for beech and 45 for fir (Table 1). The mean temperature was 9.6 °C and the mean annual precipitation 1100 mm (climate period 1973–2017). The soil parent material is characterized by sandstone with dystric cambisol and a sandy loam soil texture [55]. Climatological data were obtained from a meteorological station near the study site (<5 km). Being located at a rather low-altitude area of the Black Forest in the

sub-mountainous zone, the study area is considered to be a vulnerable growth region for beech and fir regarding increasing drought impacts under future climate change.

**Table 1.** Inventory results of the Freiamt experimental site with tree density (N/ha) and species share 65% and 35%—mean stand age and mean annual increment (MAI) (both determined with the yield table of Bösch (2001) and the measured heights of the 100 tallest trees H100).

| | European Beech | | Silver Fir | |
| --- | --- | --- | --- | --- |
| | Freiamt Site | Simulated Stand | Freiamt Site | Simulated Stand |
| **N/ha** | 404 (65%) | | 214 (35%) | |
| **N/ha** | 618 [1] (100%) | 623 (100%) | 618 [1] (100%) | 623 (100%) |
| **DBH** | 22.0 | 23.0 | 29.6 | 25.0 |
| **H100** | 23.6 | 25.0 | 24.9 | 27.0 |
| **MAI$_{100}$** | 10.0 | 10.2 | 18.0 | 17.3 |
| **age** | 55 | 50 | 45 | 50 |

[1] Calculated pure stand of beech or fir.

The photosynthetic submodule together with the leaf biomass parameters leaf mass per leaf area (LMA, g cm$^{-2}$) and leaf area index (LAI in m$^2$ m$^{-2}$) are critical parameters for the efficiency of total canopy carbon gain, and together with respiration, for forest growth and productivity. Details can be found in Supplementary Materials, Note S4. We measured LMA, the photosynthetic potential ($V_{c,max}$, $J_{max}$, TPU) and foliar night and day respiration ($R_n$ and $R_d$) in two seasonal campaigns in spring and summer 2017 for beech and for fir. A tedious three-step approach is required to obtain the photosynthetic variables: (i) Cutting twigs from the canopy with a tree-climber, (ii) immediate analysis of the leaves on the ground with a foliar gas exchange analyzer (WALZ GFS-3000) and (iii) the fitting of the generated response curves with the non-linear Farquhar-vonCaemerer-Berry (FvCB) photosynthesis model (Notes S2 in Supplementary Material). We followed a similar experimental protocol as described in [56,57], except that we conducted the gas exchange analyses on the cut twigs immediately in the field because the stomatal conductance was not as stable as for the sclerophyllous Mediterranean vegetation; see also discussion in [58]. After the photosynthesis experiments, leaves were sampled from the cut twigs and LMA was determined in the lab.

We used the Plant Canopy Analyser LAI-2200C with one below and one above canopy sensor using only the upper three rings (0–43° from zenith) [59] for monthly LAI measurements in pure stands of beech and fir from 2017 to 2019. Average LAI values of the peaks of the three years were used to parametrize the LAI module. The optical readings of the LAI-2200C represent the plant area index (PAI), including all other woody plant material (stems, branches) besides leaves. The effective LAI of beech results from the subtraction of the winter readings of the PAI during the leafy period. For evergreen conifers, the conversion from raw LAI-2200C readings to LAI is more complicated because the LAI-2200C strongly underestimates the true LAI due to needle and shoot clumping [59]. We thus multiplied the PAI with the correction factor 1.65 from *Picea abies* [60,61] due to a lack of data for Silver fir. Without a correction factor, average PAI values of fir and beech were similar (9.5 ± 0.2 and 9.7 ± 0.1, respectively).

The calibrated parameters used in the simulator were within the margin of standard error and are summarized in Table S1 (Supplementary Materials). For the soil module, we used field data from the Freiamt experimental site, including soil depth, field capacity, soil organic carbon and relative volume of stones (Table S1). This work focuses on the simulation results of the process-based model, for which reason we refer to the Supplementary Materials for further details on the experimental protocol of gas exchange and LAI measurements, photosynthesis models, etc. (Supplementary Materials, Notes 1–3).

Statistical analyses were done using a two-way ANOVA to test for differences between spring and summer, and tree species, regarding LMA, $V_{c,max}$, $J_{max}$, TPU, $R_n$ and $R_d$.

*2.3. Climate Data and Future Climate Scenarios*

Based on a climate time series (1973–2017) from a nearby meteorological station, we generated climate data for 120 years using the in-built weather generator in GOTILWA+ [62]. The variables of the climate source file were, in a daily time step, precipitation, minimum and maximum temperature, midday vapor pressure deficit, radiation, evapotranspiration, wind speed and ambient $CO_2$ concentration. The generated climate file has the same statistical structure as the seed file from the meteorological station. GOTILWA+ runs at an hourly time-step. The daily values in the meteorological file are downscaled internally by the model to the hourly values. We applied the climatic projection until 2100 generated by the MPI-ESM-LR global circulation model from the Max Planck Institute taken from the WorldClim database (http://www.worldclim.org/) (Table 2). Based on the representative concentration pathways (RCP) RCP2.6, RCP4.5 and RCP8.5 of the MPI-ESM-LR, we calculated decadal mean annual temperature increments of 1.8, 2.6 and 4.4 °C; an annual $CO_2$ increments of 0.21, 1.38 and 5.36 ppm; and total annual precipitation decrements of 25, 24 and 27 mm for the three emission scenarios (respectively). RCP2.6 is the optimistic scenario, RCP8.5 is the pessimistic scenario and RCP4.5 is the medium scenario regarding the human mitigation measures against climate change [18]. We adjusted the seasonal patterns of temperature and precipitation in the climate module of GOTILWA+ as predicted by the global circulation models. This resulted in stronger decreases of precipitation and higher peaks of temperatures in summer with more severe drought stress (Table 2).

We used the multiscalar, monthly standardized precipitation evapotranspiration index (SPEI) [63] to analyze the climate data from Freiamt and to compare the created climate data sets of the different climate scenarios no climate change (noCC), RCP2.6, RCP4.5 and RCP8.5 (R-package "SPEI" version 1.7). SPEI is a multiscalar drought index based on precipitation and also temperature. It can be used for determining the onset, duration and magnitude of drought conditions with respect to normal, average conditions.

When precipitation and potential evapotranspiration in a monthly period are lower (larger) than the average of this monthly period, conditions are drier (more humid) than normal conditions. Dry (negative SPEI values) and humid (positive SPEI values) periods are represented by red and blue bars, respectively. The monthly periods can range from 1 to 24 months. We used 3, 6, 12 and 24 months (SPEI-3, SPEI-6, SPEI-12 and SPEI-24, respectively). For instance, SPEI-3 in April is calculated by comparing SPEIs of February, March and April with the average of all SPEI in February, March and April prior to the regarded month. More details can be found online (https://spei.csic.es/index.html).

To be able to complete an entire stand development of 120 years from juvenile stage to final harvest (starting in simulation year 2000) we continued with constant values from 2100 to 2120 due to the lack of climate information from the RCP projections beyond 2100 [53]. We performed four scenarios: the reference scenario with business-as-usual management assuming no climate change (noCC) and three climate change scenarios RCP2.6, RCP4.5 and RCP8.5. For noCC, we used the generated climate data of 120 years as described above and used a constant atmospheric $CO_2$ concentration of 370 ppm (average concentration of 2000) [64]. noCC represents our reference scenario against which the effects of the RCP scenarios were compared. We defined the tipping point as when the productivity of the climate change scenarios dropped below the productivity of noCC. This accounts for when the positive effects due to $CO_2$ increase are less effective than the negative climate effects (water availability and temperature). The three RCP scenarios were additionally run with constant atmospheric $CO_2$ concentration at 370 ppm in order to investigate the extent of the "$CO_2$ fertilization" which is the positive feedback of increased atmospheric $CO_2$ concentration on vegetation growth.

**Table 2.** Definitions of 10 climate change scenarios with (a) base value for atmospheric $CO_2$ concentration ($CO_2$ Base), $CO_2$ increase in % $year^{-1}$, downregulation factor of photosynthesis (PD), increment of temperature (T increase), decrement of precipitation (P decrease) and concentration factor inducing more intense precipitation events (P factor). In (b), monthly factors of temperature increase (T in °C/100 year) and precipitation decrease (P %/100 year) are listed. Coordinates from the Freiamt site were used to get the data from the MPI-ESM-LR global circulation model from the WorldClim database (http://www.worldclim.org/).

| (a) | | | | | | |
|---|---|---|---|---|---|---|
| Scenario | $CO_2$ Base ppm | $CO_2$ Increase % $year^{-1}$ | PD % | T Increase °C/10 years | P Decrease %/100 years | P Factor - |
| noCC | 370 | 0 | 0 | 0 | 0 | 0 |
| RCP2.6 | 370 | 0.21 | 0 | 0.18 | −24.8 | 5 |
| RCP4.5 | 370 | 1.38 | 0 | 0.25 | −23.5 | 5 |
| RCP8.5 | 370 | 5.36 | 0 | 0.44 | −26.4 | 10 |
| RCP2.6-$CO_2$ | 370 | 0 | 0 | 0.18 | −24.8 | 5 |
| RCP4.5-$CO_2$ | 370 | 0 | 0 | 0.25 | −23.5 | 5 |
| RCP8.5-$CO_2$ | 370 | 0 | 0 | 0.44 | −26.4 | 10 |
| RCP8.5_PD100 | 370 | 5.36 | 100 | 0.44 | −26.4 | 10 |
| RCP8.5_PD75 | 370 | 5.36 | 75 | 0.44 | −26.4 | 10 |
| RCP8.5_PD50 | 370 | 5.36 | 50 | 0.44 | −26.4 | 10 |
| RCP8.5_PD25 | 370 | 5.36 | 25 | 0.44 | −26.4 | 10 |

| (b) | | | | | |
|---|---|---|---|---|---|
| | RCP2.6 | | RCP4.5 | | RCP8.5 | |
| Month | T in °C/100 year | P %/100 year | T in °C/100 year | P %/100 year | T in °C/100 year | P %/100 year |
| 1 | 1.61 | 14 | 2.47 | 14 | 4.04 | 16 |
| 2 | 1.01 | −22 | 1.29 | −24 | 3.29 | −13 |
| 3 | 0.35 | −13 | 0.56 | −4 | 1.64 | −4 |
| 4 | 2 | −20 | 2.07 | −4 | 3.21 | −5 |
| 5 | 1.01 | −23 | 1.87 | −32 | 3.37 | −29 |
| 6 | 2.46 | −20 | 2.96 | −15 | 5.31 | −35 |
| 7 | 2.79 | −53 | 3.86 | −63 | 6.14 | −64 |
| 8 | 1.96 | −7 | 3.11 | −19 | 6.18 | −32 |
| 9 | 3.15 | −44 | 4.01 | −47 | 6.86 | −62 |
| 10 | 1.74 | −50 | 1.96 | −38 | 4.74 | −43 |
| 11 | 2.41 | −51 | 3.26 | −39 | 5.34 | −37 |
| 12 | 1.36 | −9 | 1.94 | −11 | 3.15 | −9 |

*2.4. Management Regime*

The management module in GOTILWA+ allows thinning and regeneration of new trees in a yearly time step allowing thinning from above or below or in all DBH classes. We used the "forest development types" defined for Baden-Württemberg [65] and local silvicultural handbooks [66] to establish the management regime (see Table S2 Supplementary Materials). Thinning intensity was approximately 60 $m^3$ $ha^{-1}$ per decade for beech and approximately 80 $m^3$ $ha^{-1}$ for fir. Thinning mode in GOTILWA+ includes tree number, basal area, stem volume and biomass. We calibrated the thinning intensity in noCC with "tree volume." In the climate change scenarios, the thinning mode was "tree number" keeping the same tree density as in noCC, so that changes in productivity are due to climate effects and not due to adaptation in management. The target diameter for beech is 50–60 cm aiming at 80–100 future crop trees $ha^{-1}$ at the final harvest and 40–50 cm for fir with envisaged 200 future crop trees $ha^{-1}$.

## 2.5. Validity of Simulation Results

Our simulation results were compared with increment and yield tables for beech and fir from FVA Baden-Württemberg [67]. The mean annual incremental growth (MAI) is defined as the total accumulated growth (TAG in $m^3$ $ha^{-1}$) at a certain stand age divided by the stand age (MAI = TG/age). The MAI at stand age 100 ($MAI_{100}$ in $m^3$ $ha^{-1}$) is used to classify the productivity of stands (in German termed $dGz_{100}$—"durchschnittlicher Gesamtzuwachs"). Actual wood growth is measured with the current annual wood increment (CAI in $m^3$ $ha^{-1}$ $year^{-1}$). The data by [67] provide the CAIs of *A. alba* (a) and *F. sylvatica* (b) for different productivity classes ($MAI_{100}$ of 4, 6, 8, 10, 12, 14, 16, 18 $m^3$ $ha^{-1}$) from long-term experimental plots in Baden-Württemberg.

## 3. Results

### 3.1. Effect of Species, Season and Leaf Position on Key Ecophysiological Model Parameters

Leaf area index (LAI), leaf mass per area (LMA), the photosynthetic potential ($V_{c,max}$, $J_{max}$, TPU) and foliar night and day respiration ($R_n$ and $R_d$) are critical parameters in process-based models for upscaling foliar gas exchange to whole-canopy carbon fluxes (Table 2). LMA, $V_{c,max}$, $J_{max}$, TPU, $R_n$ and $R_d$ and were not significantly different between spring and summer tested with a two-way ANOVA ($p > 0.05$). Additionally, no significant differences were found between fir and beech in any parameter—except $J_{max}$, which was significantly higher in fir compared to beech ($p < 0.05$). Besides sunlit leaves, shaded leaves of fir were analyzed (but not of beech due to limitations in equipment and labor). Shaded leaves had significantly lower LMA, $V_{c,max}$, $J_{max}$, $R_n$ and $R_d$ compared to sunlit leaves ($p < 0.05$) (Table 3).

**Table 3.** Key ecophysiological parameters for beech and fir from the Freiamt experimental site, including photosynthetic variables from gas exchange analyses, leaf morphology and leaf area index measurements with net photosynthetic assimilation ($A_{net}$ in µmol $CO_2$ $m^{-2}$ $s^{-1}$), stomatal internal $CO_2$ concentration ($C_i$ in µmol $CO_2$ mol $air^{-1}$), chloroplastic internal $CO_2$ concentration ($C_c$ in µmol $CO_2$ mol $air^{-1}$), stomatal conductance ($g_s$ in mol $H_2O$ $m^{-2}$ $s^{-1}$), mesophyll conductance ($g_m$ in mol $m^{-2}$ $s^{-1}$ $bar^{-1}$), leaf mass per area (LMA in mg $cm^{-2}$), night and day respiration ($R_n$ and $R_d$ in µmol $CO_2$ $m^{-2}$ $s^{-1}$), maximum carboxylation capacity ($V_{c,max}$ in µmol $CO_2$ $m^{-2}$ $s^{-1}$), maximum electron transport rate ($J_{max}$ in µmol $m^{-2}$ $s^{-1}$), triose phosphate-use (TPU in µmol $m^{-2}$ $s^{-1}$) and leaf area index ($m^2$ $m^{-2}$). Only shaded leaves of fir were sampled due to limitations in equipment and labor. The table shows the apparent $V_{c,max}$, $J_{max}$ and TPU derived from $A/C_i$ curves, whereas results from $A/C_c$ curves are summarized in Table S1. Sample size was $n = 38$ per species except for gm with $n = 11$ per species. Leaf area index (LAI) values are means of the peaks from the three years 2017, 2018 and 2019.

| Tree Species | Leaf Position | $n$ | $A_{net}$ | $g_s$ | $C_i$ | $g_m$ | Cc | LMA | $R_n$ | $R_d$ | $V_{c,max}$ | $J_{max}$ | TPU | LAI |
|---|---|---|---|---|---|---|---|---|---|---|---|---|---|---|
| *F. sylcatica* | sun | 38 | 7.86 | 0.112 | 267 | 0.054 | 151 | 5.8 | 0.49 | 0.22 | 37.9 | 60.7 | 10.1 | 8.4 |
| | SE | | 0.53 | 0.010 | 7 | 0.006 | 28 | 0.3 | 0.04 | 0.03 | 2.9 | 5.2 | 2.5 | 0.2 |
| *A. alba* | Sun | 12 | 9.80 | 0.154 | 269 | 0.034 | 85 | 19.2 | 1.71 | 0.88 | 47.1 | 115.2 | 7.5 | - |
| | SE | | 1.67 | 0.027 | 9 | 0.009 | 20 | 1.6 | 0.26 | 0.14 | 6.6 | 13.4 | 0.7 | - |
| *A. alba* | shade | 12 | 5.80 | 0.066 | 214 | 0.063 | 76 | 11.3 | 2.63 | 0.29 | 31.4 | 73.9 | 4.4 | - |
| | SE | | 0.52 | 0.009 | 10 | 0.024 | 19 | 0.5 | 2.09 | 0.10 | 2.8 | 11.0 | 0.5 | - |
| *A. alba* | all | 24 | 7.80 | 0.110 | 241 | 0.047 | 81 | 15.2 | 2.17 | 0.57 | 39.3 | 94.6 | 6.0 | 14.9 |
| | SE | | 0.95 | 0.016 | 9 | 0.015 | 8 | 1.2 | 1.03 | 0.11 | 3.9 | 9.5 | 0.5 | 0.6 |

The peak of LAI was in June for both species in all three measurement years 2017, 2018 and 2019 and was on average 14.9 ± 0.6 for fir and 8.4 ± 0.2 for beech (Figure S2). The lowest LAI of the evergreen fir was in February/March (6.5 ± 0.4) (Figure S2) before the flush of new shoots. The PAI of beech in the leafless period representing woody parts only was on average 1.1 ± 0.2. When comparing the measured LAI with the mean $LAI_{sim}$ at the same day of the year (DOY), the $R^2$ is lowest for 2018 ($R^2$ of 0.40) compared to 2017 ($R^2$ of 0.70) and 2019 ($R^2$ of 0.75) (in-built scatter plots in Figure S2).

The drought year 2018 thus led to an earlier reduction of LAI starting in July. For fir, the low $R^2$ shows no good agreement in the relationship between the measured LAI and the mean $LAI_{sim}$. A mean leaf life span of 5 years was used in GOTILWA+. Shed leaves were immediately replaced with new leaves, which resulted in a much more homogenous seasonality development than indicated by the measured LAI. GOTILWA+ does not regulate the seasonality in carbon gain with the seasonality in LAI, but with the seasonality in the photosynthesis sub-module.

### 3.2. Drought Trends in the Baseline Climate Data and Climate Scenarios

Analyzing the used climate data from Freiamt with the SPEI drought index indicates a trend towards drier conditions than average in the past 15 years in the study region (Figure 2). Under the future projections for this site, this trend worsened notably for the three applied climate change scenarios (Figure S10). RCP2.6 and RCP4.5 showed comparable SPEIs, whereas in RCP8.5 the SPEI showed notably lower values and hence more severe drought conditions.

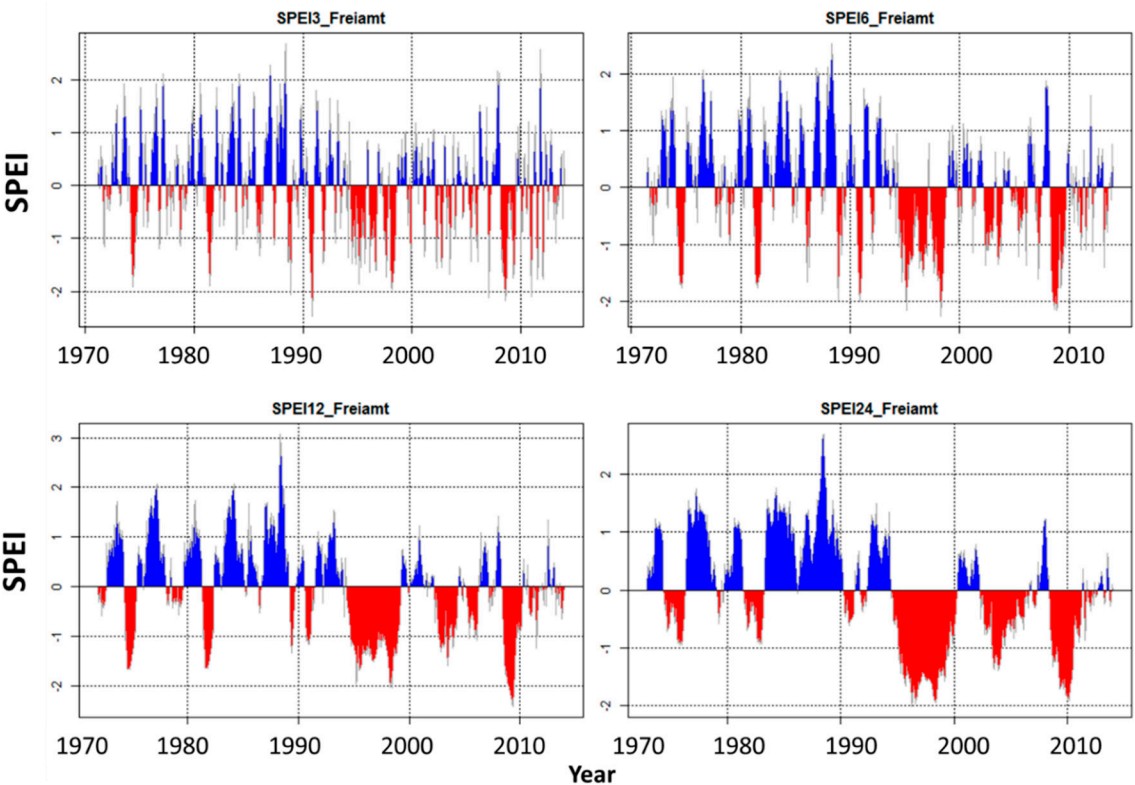

**Figure 2.** Monthly standardized precipitation evapotranspiration index (SPEI) using the climate data from a meteorological station near Freiamt (Black Forest, Germany) (1973–2017). Positive values indicate that the difference between monthly precipitation and potential evapotranspiration is larger than the average for a given monthly period. Negative values thus represent conditions drier than average. The monthly periods used were 3, 6, 12 and 24 months for SPEI-3, SPEI-6, SPEI-12 and SPEI-24, respectively.

### 3.3. Productivity of Beech and Fir in the Reference Scenario and the Climate Change Scenarios

The productivity in noCC was slightly above average compared to the medium MAI classes of beech and fir in Baden-Württemberg. Panel a1 and b1 of Figure 3 show the CAIs of *A. alba* (a) and *F. sylvatica* (b) for different productivity classes ($MAI_{100}$ of 4, 6, 8, 10, 12, 14, 16, 18 m³ ha⁻¹) summarizing stand data from long-term experimental plots from Forstliche Versuchsanstalt Baden-Württemberg (FVA) [67]. The fluctuations in productivity of the modelled stands are due to fluctuations in climate and soil water content (Figure S9, Supplementary Materials). The strong decline in CAI at stand age

105 and 110 (simulation year 2105 and 2110) was due to three years with particular low precipitation (between 767–872 mm year$^{-1}$) compared to the long-term average (1100 mm year$^{-1}$). After the calibration, the productivity of the modelled stands—MAI$_{100}$ of 10.2 m$^3$ ha$^{-1}$ for *F. sylvatica* and 17.3 m$^3$ ha$^{-1}$ for *A. alba*—met the productivity of the experimental site at Freiamt (MAI$_{100}$ of 10 and 18 m$^3$ ha$^{-1}$, Table 1). Figure 3(a2,b2) shows that CAIs of the simulated beech and fir stands are in good agreement with the CAIs of the yield table with MAI$_{100}$ values of 10 and 18 m$^3$ ha$^{-1}$. Additionally, we validated the modelled stand by comparing its standing volume and tree density with supplementary inventory plots made from the area covering a wide range of age classes (Figure S3, Supplementary Materials).

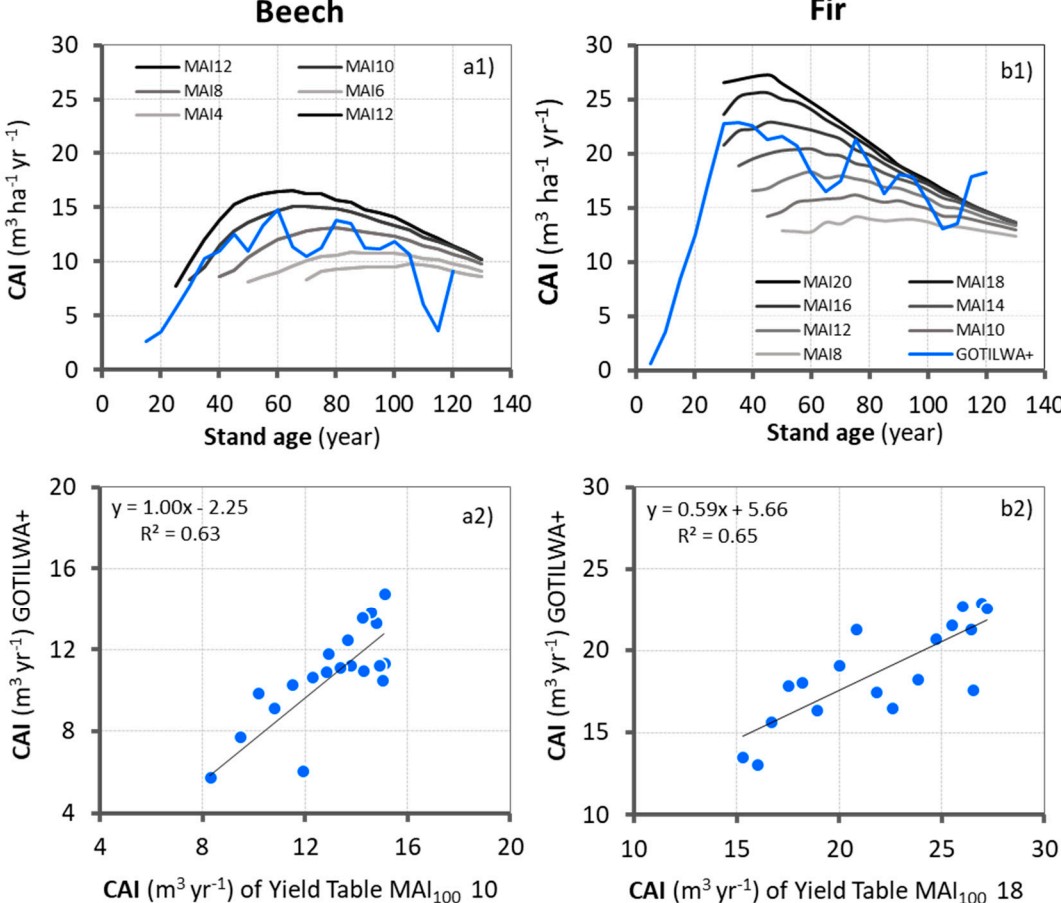

**Figure 3.** Current annual increment (CAI in m$^3$) values for (**a1**, **a2**) beech (left panel) and (**b1**, **b2**) fir (right panel) for five productivity classes from yield tables in Baden-Württemberg until stand age 130 (grey lines; data are from permanent experimental plots, FVA, 2001) and the business-as-usual scenario assuming no climate change with GOTILWA+. Productivity classes are expressed with mean annual increment (MAI in m$^3$ ha$^{-1}$) values at stand age 100 (grey lines)—for 5 and 8 productivity classes for beech and fir, respectively. Panel (**a2,b2**) show the relationship (significant at $p < 0.05$) of CAI of GOTILWA+ with the CAI of the yield table for the productivity class MAI$_{100}$ 10 for beech and MAI$_{100}$ 18 for fir.

### 3.4. Climate Change's Impacts on Net Primary Productivity

In total, net primary productivity (Mg C ha$^{-1}$ year$^{-1}$) was 16–20% and 6% lower in RCP4.5 and RCP2.6 compared to noCC, and 11–17% higher in RCP8.5 for both beech and fir (Figure 4). Characteristic of all scenarios was that initial productivity gains turned into losses (compared to noCC) after a certain tipping point. In RCP2.6, NPP was on average 6% higher until 2040 and 26% lower from 2040 to 2120 (see inset plots i1 and i2 in Figure 4a,b). In RCP4.5, NPP was on average 13.8% higher until 2045 and 17.5% lower from 2045 to 2120. In RCP8.5, NPP was on average 27.3% higher until 2060 and

9.9% lower from 2060 to 2120. This corresponds to stand ages 40, 45 and 60, as our simulations started with newly regenerated stands. Although RCP8.5 exhibited the strongest temperature increase and precipitation decrease, the positive effect of the higher $CO_2$ emissions compensated for the negative climate effects. The tipping point for RCP8.5 was thus at a later stand age than for RCP4.5 and RCP2.6. Lower values of NPP were strongly correlated with decreased soil water content (Figure S9). The length of the growth period increased 17, 20 and 34 days in RCP2.6, RCP4.5 and RCP8.5 and hence also contributed to an increased annual carbon gain.

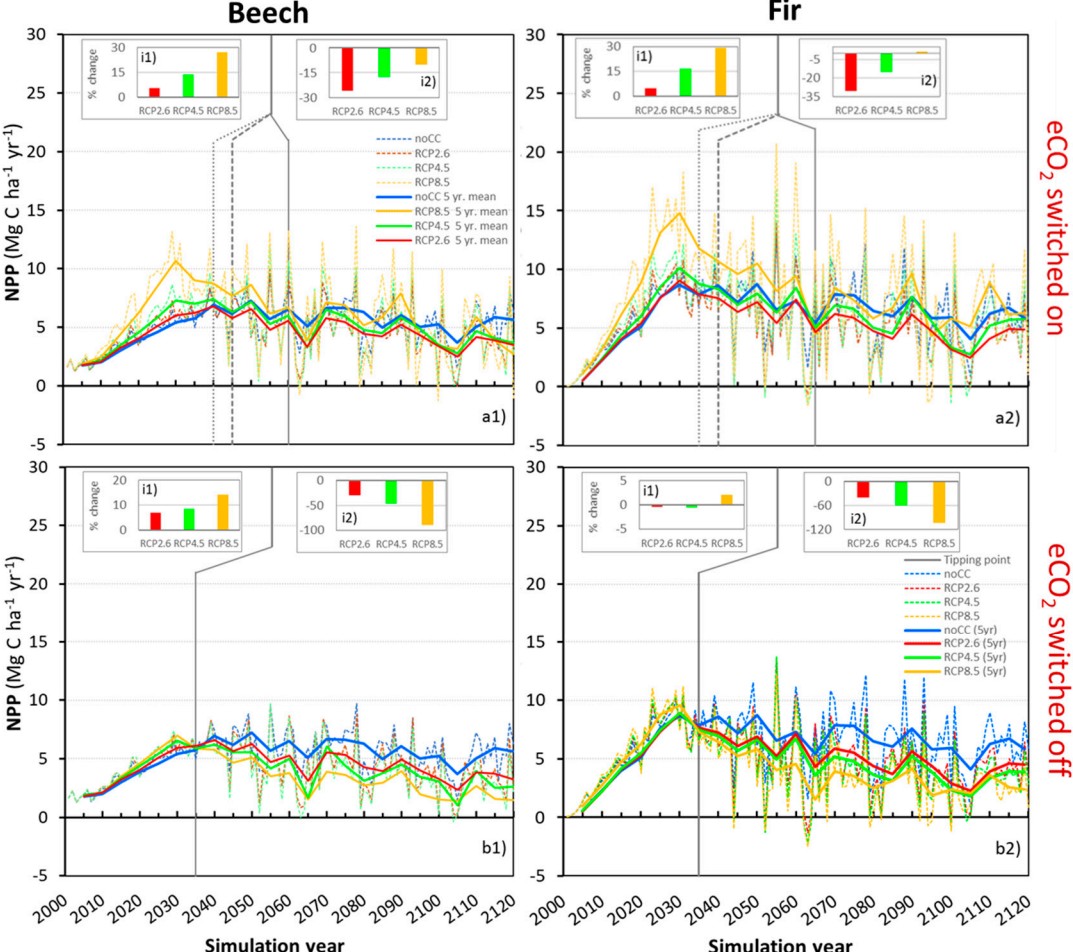

**Figure 4.** Effects of four climate scenarios (no climate change (noCC), representative concentration pathway 2.6 (RCP2.6), RCP4.5, RCP8.5) on net primary productivity (NPP) of beech (left) and fir (right) with $eCO_2$ (**a1**,**a2**) and with $eCO_2$ switched off (**b1**,**b2**). Solid lines represent the running average of 5 years. Vertical dotted, dashed and solid lines indicate the tipping points when the NPPs of the RCP scenarios fall below the reference scenario noCC. Inset plots show the differences of NPP in percent compared to noCC before and after such tipping points (i1 and i2). Simulations started with naturally regenerated juvenile stands.

### 3.5. The Effect of $CO_2$ Fertilization on Productivity

In a second simulation exercises, we disentangled the positive effect of $CO_2$ fertilization ($eCO_2$) from climate effects. RCP scenarios were repeated switching off of $eCO_2$ and applying a constant $CO_2$ concentration (e.g., RCP8.5-$CO_2$). When the $CO_2$ fertilization effect was fully disabled, RCP8.5 turned from the most productive to the least productive scenario (Figure 4). Lengthening of the vegetation period still increased and the periods with carbon gain and growth were 17, 20 and 34 days longer in RCP2.6, RCP4.5 and RCP8.5, respectively (data not shown). This led to higher NPPs compared to noCC, until the tipping point at simulation year 2035 when the positive effects were overtaken by the

negative climate effects (Figure 4). Before the tipping point, NPP was 0–14% higher, and after the tipping point 30–104% lower than in noCC (both species). Averaged over the entire simulation period, NPP was 20–54% lower than in noCC.

### 3.6. Accounting for Photosynthetic Downregulation as a Response to eCO$_2$

In a third simulation exercise, we applied photosynthetic downregulation (PD) combined with eCO$_2$ (e.g., RCP8.5-PD) to account for the fact that plants may benefit from eCO$_2$, but may not fully benefit from the fertilization effect due to nutrients and/or plant internal acclimation responses. Four levels of PD with 25, 50, 75 and 100% were chosen because the photosynthetic machinery may acclimate at varying degrees to eCO$_2$. Increasing levels of PD gradually cancelled out the positive feedback of NPP to eCO$_2$ (Figure 5). At full PD of 100% (RCP8.5-PD100), NPP fell to a similar level to at constant CO$_2$ (RCP8.5-CO$_2$), but was still 4–5% higher (Figure 5). Additionally SV was strongly reduced with eCO$_2$ switched off and decreased approximately 50% by the end of the rotation (Figure 6).

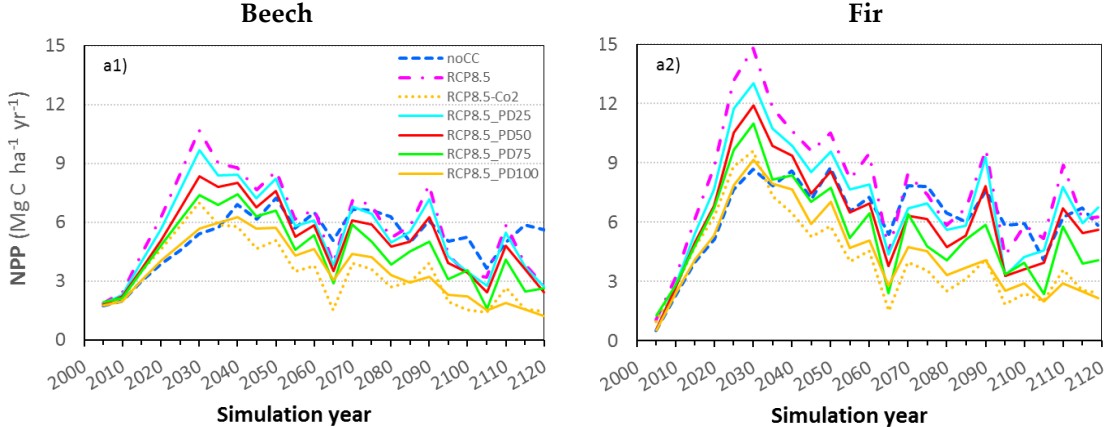

**Figure 5.** Effects of the climate change scenario RCP8.5 on net primary productivity (NPP) of beech (**a1**) and fir (**a2**) with constant CO$_2$ concentration (370 ppm—global mean of 2000) (RCP8.5-CO$_2$) and with increasing CO$_2$ concentration assuming photosynthetic downregulation by 25, 50, 75 and 100% (RCP8.5_PD25, RCP8.5_PD50, RCP8.5_PD75, RCP8.5_PD100). Simulations started with a naturally regenerated juvenile stand.

### 3.7. Climate Change's Impacts on Standing Volume, Basal Area and Mortality

Standing volume (SV) and basal area (BA) were notably above noCC in RCP8.5. In RCP2.6 and RCP4.5, SV and BA followed similar trends as in noCC. SV of beech and fir then dropped below noCC in 2055 and 2045 for RCP2.6 and in 2070 and 2060 for RCP4.5, respectively (Figure 6). In RCP8.5, the SV of both species did not fall below noCC, but peaked at 499 and 526 m$^3$ ha$^{-1}$ in 2060 (respectively) and then equaled the value in noCC towards the end of the simulation period. All three scenarios showed a declining trend after 2060. RCP2.6 accumulated the lowest standing volume for beech and was 100 m$^3$ ha$^{-1}$ lower than noCC at the end of rotation. Compared to NPP, eCO$_2$ had a more persistent impact on SV due to the accumulation of carbon in standing timber. The accumulation of SV in the RCP scenarios also led to a higher harvesting volume (HV) for beech until 2060 in RCP2.6 and 4.6, and in RCP8.5 until 2100 (Figure S4, Supplementary Materials). For Fir, the HV was higher until 2030 in RCP2.6, until 2055 in RCP4.5 and until 2110 in RCP8.5 (Figure S4, Supplementary Materials). In total, the accumulated HV of the entire simulation was 13–19% lower in RCP2.6, 1–9% lower in RCP4.5 and 21–29% higher in RCP8.5. The number of harvested trees was identical in all scenarios so that the change in HV with respect to noCC was due to changes in tree growth.

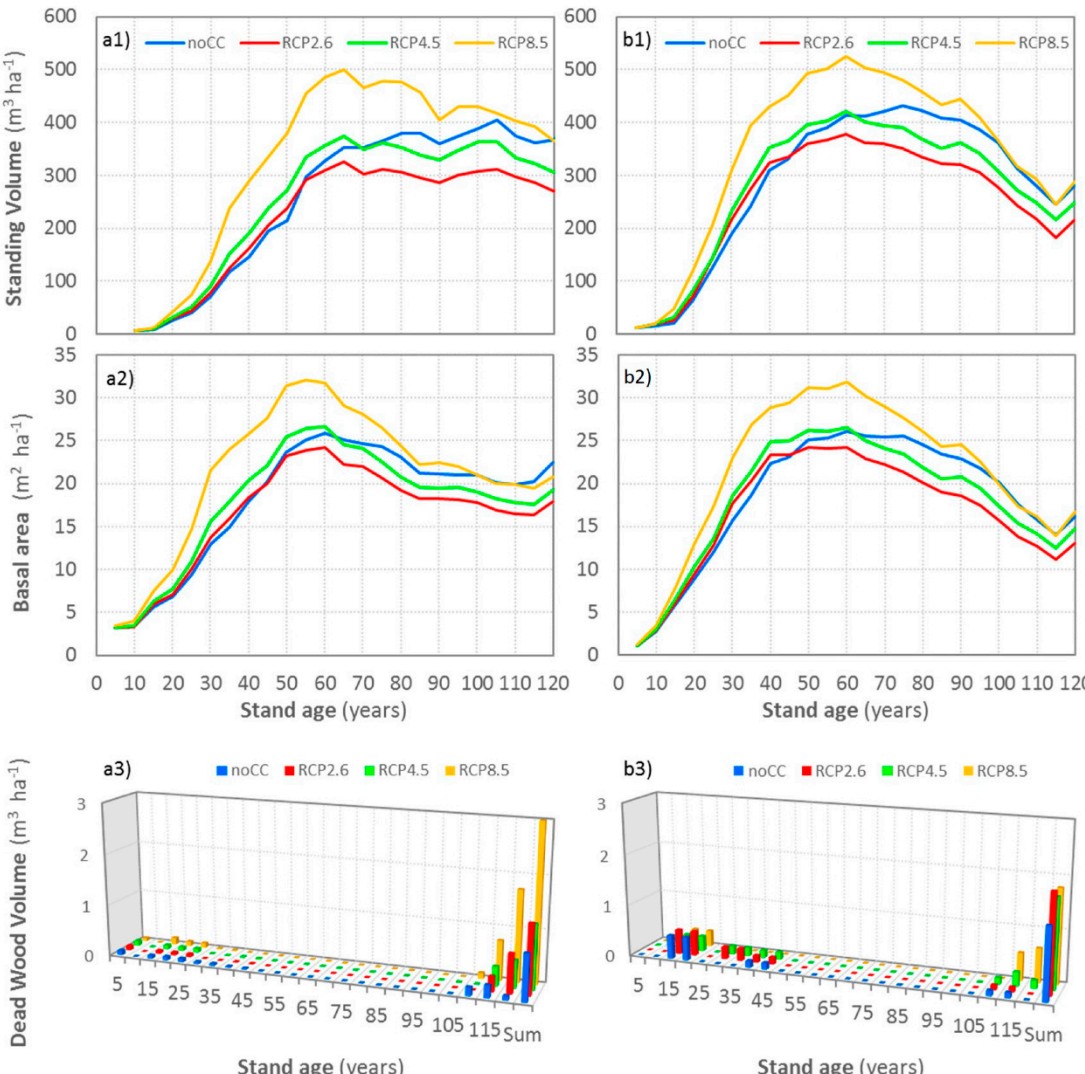

**Figure 6.** Effects of four climate scenarios (noCC, RCP2.6, RCP4.5, RCP8.5) on standing volume (1), basal area (2) and dead wood volume (3) of beech (**a1**–**a3**) and fir (**b1**–**b3**). For the noCC scenario, a climate file of 120 was generated with an in-built weather generator in GOTILWA+ using climate data of the past 40 years from a nearby meteorological weather station with constant $CO_2$ concentration (370 ppm—global mean of 2018). For the three climate change scenarios RCP2.6, RCP4.5 and RCP8.5, changes in temperature, precipitation and $CO_2$ were applied to the generated climate file according to the MPI-ESM-LR global circulation model.

More extreme climatic conditions increased the risk of drought-induced mortality. The accumulated dead wood volume (DWV) at the end of the simulation period was higher for beech than for fir. DWV in RCP2.6, RCP4.5 and RCP8.5 were 2, 1.5 and 4.5 times higher for beech and 1.4, 1.2 and 1.4 times higher compared to noCC, respectively (Figure 6). Switching off $eCO_2$ further increased DWV (Supplementary Materials Figure S5). For beach, DWV increased 17, 22 and 30% in RCP2.6-$CO_2$, RCP4.5-$CO_2$ and RCP8.5-$CO_2$ compared to RCP2.6, RCP4.5 and RCP8.5 (respectively); for fir the results were 1, 17 and −68%. The scenario RCP8.5_PD100 resulted in slightly lower DWV (2%) than RCP8.5-$CO_2$ (data not shown). Despite the DWV increase, productivity was not notably decreased.

*3.8. The $CO_2$ Fertilization and Climate Effects on Water-Use Efficiency*

The water-use efficiency (WUE) is a good indicator for the capability of plants to cope with drought stress. The applied scenarios had diverging effects on WUE through the interplay of changes

of temperature, precipitation and $CO_2$. The decreasing water availability reduced both transpiration water loss and WUE in RCP2.6 (Figure S6, Supplementary Materials). In RCP4.5, WUE was thus higher than in RCP2.6 and also noCC (Figure S6, Supplementary Materials). Although precipitation was lowest in RCP8.5, WUE was higher than all other scenarios due to the high levels of $CO_2$ (Figure S6, Supplementary Materials). With $eCO_2$ switched off, WUE reached the lowest value in RCP8.5-$CO_2$ (Figure 7). Increasing PD reduced the contribution of $eCO_2$ and thus WUE (Figure 7). WUE of RCP8.5-PD100 was, however, still well above both RCP8.5-$CO_2$ and noCC because $eCO_2$ reduced transpiration, which resulted in 4–5% higher NPP RCP8.5-PD100 compared to RCP8.5-$CO_2$ (Figure 5). In scenarios RCP2.6-PD100 and RCP4.5-PD100, however, WUE was lower than in noCC (Supplementary Materials, Figure S8) because of a lower $CO_2$ increase compared to RCP8.5-PD100, which compensated less effectively drought effects.

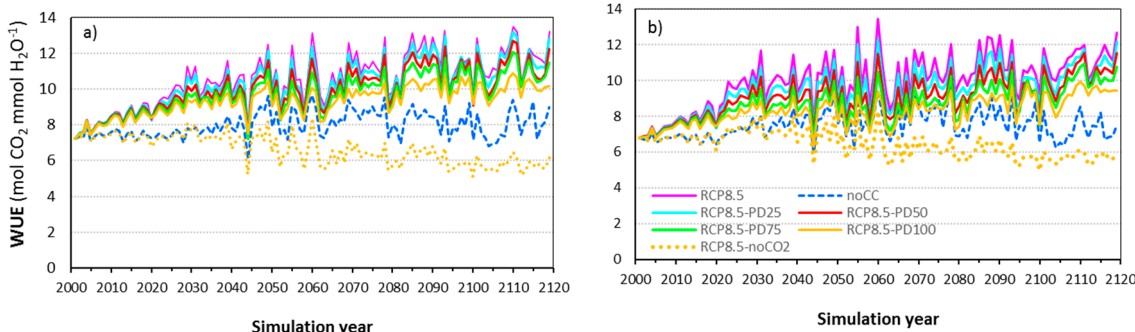

**Figure 7.** Effects of the climate change scenario RCP8.5 with constant $CO_2$ concentration (370 ppm—global mean of 2018) (RCP8.5-$CO_2$) and with increasing $CO_2$ concentration assuming photosynthetic downregulation by 25, 50, 75 and 100% (RCP8.5_PD25, RCP8.5_PD50, RCP8.5_PD75, RCP8.5_PD100) on water use efficiency (WUE) of beech (**a**) and fir (**b**). The climate scenario noCC is displayed for comparison assuming no change in precipitation and temperature and constant $CO_2$ concentration. Simulations started with a naturally regenerated juvenile stand.

## 4. Discussion

This study aimed at disentangling the effects of climate and $CO_2$ fertilization on growth processes and productivity in a potential drought risk area in the sub-mountainous belt of the Black Forest (440 m). We explicitly included the possibility for photosynthetic downregulation due to the uncertainty related to the $CO_2$ fertilization effect. This study is timely and important because forest managers have to deal with the ecological uncertainties regarding species occurrence, growth and mortality under future climate in the decision-making process.

### 4.1. Productivity Gains or Losses? The Uncertainty of the $CO_2$ Fertilization Effect

This study projects a notable stimulation of NPP for both beech and fir by elevated atmospheric $CO_2$ ($eCO_2$) with respect to our reference scenario until 2035–2060. This stimulation was, however, gradually cancelled out by climate effects until the end of simulation period due to the temperature increase and precipitation decrease. Experimental evidence confirms that, in the absence of nutrient limitations, $eCO_2$ increases productivity, but also that this effect may diminish over time [6,16,27,28]. The turning point after which NPP fell below the reference scenario, and the extent of productivity gain/loss before/after this turning point were scenario-dependent, varying between simulation years 2040 and 2060 (stand age 40–60). NPP increased on average 6–27% or 5–29% up to the turning point and decreased thereafter 10–26% or 0–30% until end of the simulation period (respectively) (Figure 4). Although both species responded similarly, fir benefitted more from $eCO_2$, confirming ecological theory [68].

Common for both species was that, in scenario RCP8.5, the turning point occurred later, the productivity gains were higher and losses were lower than for the other scenarios. These results

seem counterintuitive, as we presented RCP8.5 as the pessimistic, RCP2.6 as the optimistic and RCP4.6 as the medium scenario. Nonetheless, scenario RCP8.5 is to date in closest agreement with global atmospheric $CO_2$ emissions [69]. Although RCP8.5 projects the most extreme climate, it also projects the highest $CO_2$ emissions which compensated for climate-induced productivity losses. Switching off this compensatory effect turned the most productive RCP8.5 into the scenario with the biggest losses and the least productive RCP2.6 into the scenario with the lowest losses (on average 33–71% and 26–51%, respectively). Despite switching off e$CO_2$, initial productivity gains were between 3–14% in the first 35–40 simulation years. Rising temperatures can lengthen the growing season—reportedly 11 days in Europe since the sixties—and enhance productivity [70,71]. In our study, the growing season was extended by 17, 20 and 34 days in RCP2.6, RCP4.5 and RCP8.5, although its effect on carbon gain was neglectable compared to the $CO_2$ fertilization effect. The observed trends of advanced leaf unfolding and delayed leaf coloring appear to have decelerated and reversed in recent years [72]. Compared to NPP, e$CO_2$ had a more persistent impact on SV due to the accumulation of carbon in standing timber, which also increased the harvesting volume. However, all three scenarios showed a declining trend in SV after 2060. RCP2.6 accumulated the lowest standing volume for beech, and was 100 m$^3$ ha$^{-1}$ lower than noCC at the end of rotation.

Although we found that the positive growth stimulation was cancelled out by the middle of this century (even under e$CO_2$), for most European regions notable increases in NPP are projected by the end of this century [6,21,73–75]. In [74], NPP increased in 6 out of 10 environmental zones in Europe, even if e$CO_2$ was not considered. The reason for this discrepancy is most probably either the different implementations of water limitations and the coupling between C uptake and transpiration—and not photosynthesis, as most models have adopted similar representations of photosynthetic processes [73], or the positive effect from rising temperatures in photosynthesis kinetics in cold regions. In GOTILWA+, increasing soil water deficit is mechanistically represented because it directly affects the photosynthetic kinetics—i.e., drought-induced biochemical limitations of photosynthesis ([51,76] and see Figure S9, Supplementary Materials).

The role of e$CO_2$ as a fertilizer for plant growth has generated ongoing discussion since the very beginning of the development of dynamic global vegetation models [22,77–80]. While it is questionable to keep the $CO_2$ fertilization effect switched on without any limitation or acclimation, it is in the same vein disputable to assume no productivity gains at e$CO_2$. Nitrogen availability and C–N interactions can constrain the stimulation of NPP with e$CO_2$, but the extent of the limitation is uncertain [22,80–82]. Additionally, while e$CO_2$ certainly impacts WUE, increased leaf area may counterbalance net water savings [83] just like an increased atmospheric evaporative demand in a drier, warmer future [84].

In our study, photosynthesis had to be downregulated by more than 30% to compensate for NPP gains in RCP8.5 and to equalize the NPP of noCC. Determining a realistic downregulation factor is complicated because it depends on species-specific, plant internal acclimation mechanisms. Most studies disregard the effect of improved WUE on NPP under e$CO_2$ [33,73], which reduced the loss in NPP up to 4–5% in our study. This amount may seem small, but can be important to mitigate drought-induced mortality, because it may maintain soil water potential above lethal values for a longer period, thereby helping to withstand drought periods [33–35], even if no overall water savings are observed at a yearly time-scale [83].

We underline the importance of representing detailed soil processes and to mechanistically link water availability and water-use efficiency with photosynthesis in forest growth models. We also stress that unaccounted photosynthetic downregulation due to potential acclimation and/or nutrient limitations likely overestimates and switching off e$CO_2$ underestimates the $CO_2$ fertilization effect. The magnitude is still highly uncertain and may be different in other significant biomes than temperate (e.g., boreal or tropical), which are often underrepresented in experimental studies [85]. Including e$CO_2$ in the simulation was a real game changer and stresses the need to intensify efforts to investigate the effects of $CO_2$ fertilization, potential acclimation and improved WUE in combined field and modelling approaches.

*4.2. Climate-Driven Mortality Increased, but at a Low Rate*

Forest disturbances and mortality are globally increasing and are predicted to continue to rise as a response to climatic change [8,53,86–88]. Tree mortality sub-models play a dominant role in long-term forest dynamics and are yet still highly uncertain [53,87]. Mortality formulations empirical in nature, based on inventory data or the self-thinning law [89,90], cannot capture the climate-carbon feedback on vegetation dynamics and mortality [87,91]. GOTILWA+, however, uses the carbon balance approach and calculates loss of tree hydraulic conductance based on an annual supply (transpiration) to demand (evaporative demand) approach. Both processes are key to mechanistically represent the processes of mortality under drought [92].

We found that climate-driven mortality increased depending on timing, species and severity of climate change. RCP8.5 was characterized by the highest productivity but also by the highest mortality, and vice versa for RCP2.6. Switching off $eCO_2$ increased the total deadwood volume (DWV) between 17 and 30%. Mortality peaks were generally at the beginning due to the tough competition in young stands and in mature stands at the end of rotation, affecting regeneration but also harvestable trees. DWV was four times higher for beech, suggesting fir to be more drought tolerant. Regional observations support this finding: crown defoliation and mortality increased for beech but not for fir after the drought year 2018 [93]. However, drought predisposition makes fir vulnerable to bark beetle calamities, which increased the mortality rate threefold in 2019 [45]. Other disturbance regimes besides the climate should thus be accounted for in modelling forest productivity [94], which could have—if included—increases the DWV in our study.

To conclude, all our scenarios point towards a mortality increase for both beech and fir and a dampening effect by $eCO_2$. Although this increase was too small to notably affect forest productivity, the timing of mortality is critical. Great economic losses can be generated when losing trees at the most profitable harvestable age or through costly replanting measures due to regeneration failures.

## 5. Conclusions

There exists a great deal of uncertainty regarding how growth dynamics and mortality will unfold in the future depending on the severity of climate change, the $CO_2$ fertilization effect and the responses by forest ecosystems. Assuming a full $CO_2$ fertilization effect, negative climate impacts were compensated for and productivity was even increased—but only until a certain tipping point after which productivity decreased towards the end of the century. The lower the positive impact by $CO_2$ fertilization, the earlier the tipping point and the greater the losses. The increases in standing and harvesting volume dropped below the reference scenario after a tipping point. This will require adaptation in forest management plans and timber yields. We stress that we have not included other biotic or abiotic risks—for instance, windthrow, pests, diseases, etc—besides climatic drought, which may increase risks for forest management besides drought and heat spells. Forest areas with less drought risk in the Black Forest—for instance, at higher altitudes—may witness net gains in productivity due to $eCO_2$ and increasing growing season length, although this was not addressed in this study. Our study shows that forest management with beech and fir may still be possible under the future climate in potential risk areas such as the sub-mountainous belt of the Black Forest, but major losses in productivity and increased mortality can be expected. Under the most pessimistic scenario, forest management might approach a critical threshold where a substantial transition on the ecological and economic levels will become inevitable. Whether this means that forest management will become unprofitable with these two species under the future climate conditions is, however, another side of the coin that requires a more detailed economic analyses, for which this study provides the necessary data base.

**Supplementary Materials:** The following are available online at http://www.mdpi.com/2225-1154/8/12/141/s1, Figure S1. Schematic overview of GOTILWA+ of the ecophysiological and bio-geochemical growth module (a) and the forest management module (b). More details on each module can be found online by zooming in each module of the dynamic scheme available at https://prezi.com/to-nd8yjmbaa/gotilwa-a-process-based-forest-growth-model;

Figure S2. Measured leaf area index (LAI in $m^2 m^{-2}$) of beech (a) and fir (b) at Freiamt at the day of the year (DOY) 2017, 2018 and 2019 and simulated LAI in GOTILWA+ at stand age 40 to 60 years (GOTILWA 40-60) (age range of the experimental forest) for (a) beech and (b) fir. In-built scatter plots show the regression equation and R2 of measured (LAImea) versus simulated LAI (LAIsim). LAIsim is calculated as the mean from the 20 years for stand age 40 to 60 at the same DOY as for LAImea; Figure S3. Standing wood volume (a) (SV, overbark in m3 ha-1) and tree density (b) (N) of modelled stands and of inventory plots nearby the Freiamt experimental site; Figure S4. Effect of four climate scenarios (noCC, RCP2.6, RCP4.5, RCP8.5) on total accumulated growth (TAG) (1) and current annual increment (CAI) (2) and harvesting volume (HV) of beech (a) and fir (b). For the noCC scenario, a climate file of 120 was generated with an in-built weather generator in GOTILWA+ using climate data of the past 40 years from a nearby meteorological weather station with constant $CO_2$ concentration (370 ppm—global mean of 2018). For the three climate change scenarios RCP2.6, RCP4.5 and RCP8.5, changes in temperature, precipitation and CO2 were applied to the generated climate file—according to the MPI-ESM-LR global circulation model; Figure S5. Effect of the climate change scenario RCP8.5 with constant $CO_2$ concentration (370 ppm—global mean of 2018) (RCP8.5-$CO_2$) and with increasing CO2 concentration assuming photosynthetic downregulation by 25, 50, 75 and 100% (RCP8.5_PD25, RCP8.5_PD50, RCP8.5_PD75, RCP8.5_PD100) on standing volume (1), basal area (2), and dead wood volume (3) of beech (a) and fir (b). The climate scenario noCC is displayed for comparison assuming no change in precipitation and temperature and constant $CO_2$ concentration. The climate data was generated with an in-built weather generator in GOTILWA+ with climate data of the past 43 years from a nearby meteorological weather station. The simulations started with juvenile forests (stand age 0), which corresponds to simulation year 2000; Figure S6. Effect of four climate scenarios (noCC, RCP2.6, RCP4.5, RCP8.5) on water-use efficiency of beech (a) and fir (b). For the noCC scenario, a climate file of 120 was generated with an in-built weather generator in GOTILWA+ using climate data of the past 40 years from a nearby meteorological weather station with constant $CO_2$ concentration (370 ppm—global mean of 2018). For the three climate change scenarios RCP2.6, RCP4.5 and RCP8.5, changes in temperature, precipitation and $CO_2$ were applied to the generated climate file according to the MPI-ESM-LR global circulation model. The simulations started with juvenile forests (stand age 0), which corresponds to simulation year 2000; Figure S7. Effect of four climate scenarios (noCC, RCP2.6, RCP4.5, RCP8.5) with constant $CO_2$ concentration (370 ppm—global mean of 2018) on water-use efficiency of beech (a) and fir (b). For the noCC scenario, a climate file of 120 was generated with an in-built weather generator in GOTILWA+ using climate data of the past 40 years from a nearby meteorological weather station. For the three climate change scenarios RCP2.6, RCP4.5 and RCP8.5, changes in temperature, precipitation and CO2 were applied to the generated climate file according to the MPI-ESM-LR global circulation model. The simulations started with juvenile forests (stand age 0), which corresponds to simulation year 2000; Figure S8. Four climate scenarios (noCC, RCP2.6, RCP4.5, RCP8.5) with increasing $CO_2$ concentration on water-use efficiency with 100% photosynthetic downregulation of beech (a) and fir (b). The climate scenario noCC is displayed for comparison assuming no change in precipitation and temperature and constant $CO_2$ concentration. The climate data was generated with an in-built weather generator in GOTILWA+ with climate data of the past 40 years from a nearby meteorological weather station. The simulations started with juvenile forests (stand age 0), which corresponds to simulation year 2000; Figure S9. Relationship of net primary productivity (NPP) with soil water content (SWC) for beech and for fir for three scenarios no climate change (reference scenario) and RCP8.5 with constant $CO_2$ (370 ppm) and RCP8.5 with photosynthetic downregulation of 100% (PD100); Figure S10. Monthly Standardized Precipitation Evapotranspiration Index (SPEI) of the climate scenarios no climate change (noCC), RCP2.6, RCP4.5 and RCP8.5. Positive values indicate that the difference between monthly precipitation and potential evapotranspiration is larger than the average for a given monthly period. Negative values thus represent conditions drier than average. The monthly periods used were 3, 6, 12 and 24 months for SPEI-3, SPEI-6, SPEI-6 and SPEI-24, respectively; Table S1. Parameters of beech and fir used for different submodules (a-g) in GOTILWA+. Reference indicates the source of the used parameter originating from a pre-setting of GOTILWA+ (GOT), the Freiamt experimental site (FRA), measured parameter (meas), calibrated parameter of a pre-setting of GOTILWA+ (cal), setting by the user (user). For alometric relationships and wood density in (e) following references were used [1–6]; Table S2. Table displaying management interventions in GOTILWA+ for beech (a) and fir (b)with the year of intervention, the DBH class of intervention (small, big or all DBH classes), the mode of thinning (trees, basal area, standing volume, or biomass), the intensity of thinning (positive signs indicated the number of thinned trees and negative signs the tree number of the remaining stand after thinning), number of regenerated trees (regeneration), and the total tree number of the stand. Interventions are every five years except for the initialisation period (first 35 years). During the initialisation period a diameter distribution was created calibrated with inventory data from Freiamt; Table S3. Natural data per ha of business-as-usual simulations (noCC) at 5 year cycles for beech (a) and fir (b) displaying the tree number (N), standing volume (over bark), harvesting volume (over bark), diameter at breast height (DBH), height (H), basal area (BA), current annual increment (CAI), mean annual increment (MAI), total accumulated growth (TAG), total biomass (TBM, above- and belowground), deadwood volume (DWV), and number of dead trees (mortality); Table S4. Time table displaying stem density (N), standing volume (SV in $m^3 ha^{-1}$), harvesting volume (HV in $m^3 ha^{-1}$), total accumulated growth (TAG in $m^3 ha^{-1}$), diameter at breast height (DBH in cm), tree height (H in m) and basal area (BA in $m^2$) of one rotation length of beech (a) and fir (b) simulated with GOTILWA+ assuming no climate change (noCC), climate change with RCP2.6, RCP 4.0 and RCP8.5 (changes in temperature, precipitation and $CO_2$ as in Table S1). The thinning intensity for the three climate change scenarios was applied via stem number reductions keeping the same tree density at each interval as for the noCC scenario. Age of the stand was zero at the start of the simulation corresponding to year 2000; Note S1:

Leaf Sampling and leaf morphology; Note S2: Photosynthetic gas exchange measurements—$CO_2$-response curves; Note S3: LAI measurements; Note S4: GOTILWA+: productivity, drought and mortality.

**Author Contributions:** Conceptualization: D.S., M.H. and R.Y.; methodology: D.S., D.N.-S.; C.G., J.K., M.K., R.Y. software: D.S., D.N.-S., C.G.; validation: D.S., M.K. and R.Y.; formal analysis: D.S., D.N.-S., R.Y.; investigation: D.S., D.N.-S.; C.G., J.K., M.K., R.Y.; resources: D.S., C.G., J.K.; writing—original draft preparation: D.S.; writing—review and editing: D.S., D.N.-S.; C.G., J.K., M.K., R.Y.; visualization: D.S.; supervision: M.K. and R.Y.; funding acquisition: M.K. and R.Y. All authors have read and agreed to the published version of the manuscript.

**Funding:** The present study is part of the project "Buchen-Tannen-Mischwälder zur Anpassung von Wirtschaftswäldern an Extremereignisse des Klimawandels (BuTaKli)" within the program "Waldklimafonds" (number 22WC106901) which was financially supported by the Bundesministerium für Ernährung und Landwirtschaft (BMEL), the Bundesministerium für Umwelt, Naturschutz, the Federal Minister of Agriculture (BML) and the Federal Minister of Environment (BMU) via the Federal Institute of Agriculture and Nutrition (BLE) (grant number FKZ/28W-C-1-069-01). The article processing charge was funded by the Baden-Wuerttemberg Ministry of Science, Research and Art and the University of Freiburg in the funding programme Open Access Publishing.

**Acknowledgments:** Inventory data from the experimental site were kindly provided by Julia Schwarz from the Chair of Silviculture at the University of Freiburg. We sincerely thank Tim Burzlaff from the Chair of Forest Zoology and Entomology at the University of Freiburg for his assistance and support in field work and coordination of the project BuTaKli. We sincerely thank Raphael Trautmann for his valuable assistance in the sampling of the ecophysiological data.

**Conflicts of Interest:** The authors declare no conflict of interest.

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
