# Peer review of "Gains or Losses in Forest Productivity under Climate Change? The Uncertainty of CO2 Fertilization and Climate Effects"

_climate, doi:10.3390/cli8120141_

Round 1
Reviewer 1 Report
Dear authors,
First, I would like to thank you for presenting your results. Overall, I found that the work is interesting. In my opinion the research was done well. The GOTILWA + model is a very interesting research tool. The article is understandably written and well-organised. The manuscript itself, however, needs improvement, especially section Methods & Materials. The ecophysiological dataset is adequately described, but the climatic input data is unclear.
(see line-by-line comments below).
Major comments on text:
In the section Materials and Methods, 2.3. Climate data & Future Climate scenarios, I would suggest adding information about the climatic data and SPEI.
- Lines 186-187: “Based on a climate time series (1973-2017) from a nearby meteorological station, we generated climate data for 120 years using the in-built weather generator in GOTILWA+ “
I have a question: What climate variables were used as input datasets; whether all the data provided by the model were used, i.e .: tasmax, tas, tasmin, rlds, wind, RHS, RSDS, ps, pr
- I think you should explain a little more about SPEI (lines 180-187).
We used the multiscalar, monthly Standardized Precipitation Evapotranspiration Index (SPEI) (Vicente-Serrano et al., 2010) to analyse the climate data from Freiamt and to compare the created climate data sets of the different climate scenarios noCC, RCP2.6, RCP4.5 and RCP8.5 (R-package ‘SPEI’ version 1.7). The drought assessment by this index includes the effects of both precipitation and temperature. Positive values indicate that the difference between monthly precipitation and potential evapotranspiration is larger than the average of the used 43-year climate base for a given monthly period. Negative values thus represent conditions drier than average.
What did you mean by that?
- I think that SPEI information contained in these lines are not very clear and it is necessary to complete it, for example, on SPEI at different time scales. You give information about SPEI 3, SPEI 6, SPEI 12, SPEI 24 in section Results (lines 264-268, Figure 1), but there is no such information in the section Methods & Material. I think that information about the SPEI at different scales should be included in the Methods & Material section
- Were the SPEI values ​​also input to the model?
- Line 186 - is larger than the average of the used 43-year climate base”: can you explain?
Conclusion
Lines 498-499: We stress that we have not included other biotic or abiotic risks beside climate (for instance windthrown, pest, diseases etc.), which may increase risks for forest management beside drought and heat spells. –
What did you mean by that?
Heat spells = heat waves?
Windthrown is generally caused by wind, so it seems to be a climate risk too.
Minor comments:
Line 278 – “and fir in Baden-Württemberg. Fig. 1 shows the CAI of A. alba (a) and F. sylvatica (b) for different” – should be Fig 2.
Figure 3 – Fir - no description for RP8 simulation
Lines 188-189
Table 2: Definition of 10 climate change scenarios with a) base value for atmospheric CO2 concentration (CO2 Base), CO2 increase in % year-1, downregulation factor of photosynthesis, - should be abbreviation, I suppose: PD, increment of temperature (T increase), decrement of precipitation (P decrease) and concentration
Best regards,
Author Response
Response to Reviewer 1
Reply by authors: Thank you very much for your positive, constructive feedback and your time dedicated to review our manuscript. We have tried to address all of your comments and hopefully increased clarity in the material and methods regarding SPEI and climate input data. The figure order has been updated and some figures have been improved. Find our detailed point-by-point responses below.
Reviewer 1: Major comments on text:
In the section Materials and Methods, 2.3. Climate data & Future Climate scenarios, I would suggest adding information about the climatic data and SPEI.
- Lines 186-187: “Based on a climate time series (1973-2017) from a nearby meteorological station, we generated climate data for 120 years using the in-built weather generator in GOTILWA+ “
I have a question: What climate variables were used as input datasets; whether all the data provided by the model were used, i.e .: tasmax, tas, tasmin, rlds, wind, RHS, RSDS, ps, pr
Reply by authors: The variables of the climate source file were, in a daily time step, precipitation, minimum and maximum temperature, midday vapour pressure deficit, radiation, evapotranspiration, wind speed, and ambient CO2- concentration Such variables were down-scaled internally by GOTILWA+ to run at hourly time step. These hourly values were the ones used in the model. We have added this information in L. 176-182.
- I think you should explain a little more about SPEI (lines 180-187).
We used the multiscalar, monthly Standardized Precipitation Evapotranspiration Index (SPEI) (Vicente-Serrano et al., 2010) to analyse the climate data from Freiamt and to compare the created climate data sets of the different climate scenarios noCC, RCP2.6, RCP4.5 and RCP8.5 (R-package ‘SPEI’ version 1.7). The drought assessment by this index includes the effects of both precipitation and temperature. Positive values indicate that the difference between monthly precipitation and potential evapotranspiration is larger than the average of the used 43-year climate base for a given monthly period. Negative values thus represent conditions drier than average.
What did you mean by that?
Reply by authors: The SPEI is a multi-scalar drought index based on climatic data. It can be used for determining the onset, duration and magnitude of drought conditions with respect to normal conditions in a variety of natural and managed systems such as crops, ecosystems, rivers, water resources, etc. Dry (negative SPEI values) and humid (positive SPEI values) periods are represented by red and blue bars, respectively. We have now clarified the text and gave an example for illustration (l.194-207). More information can be found online (https://spei.csic.es/) or here (Vicente-Serrano et al., 2010).
- I think that SPEI information contained in these lines are not very clear and it is necessary to complete it, for example, on SPEI at different time scales. You give information about SPEI 3, SPEI 6, SPEI 12, SPEI 24 in section Results (lines 264-268, Figure 1), but there is no such information in the section Methods & Material. I think that information about the SPEI at different scales should be included in the Methods & Material section
Reply by authors: Thank you for making us aware of this omission. We have now explained this in the text L. 194-207
- Were the SPEI values ​​also input to the model?
Reply by authors: No, they were not input to the model. We used the generated climate file of the model to calculate SPEI afterwards.
- Line 186 - is larger than the average of the used 43-year climate base”: can you explain?
Reply by authors: We have addressed this question above.
Conclusion
Lines 498-499: We stress that we have not included other biotic or abiotic risks beside climate (for instance windthrown, pest, diseases etc.), which may increase risks for forest management beside drought and heat spells. –
What did you mean by that?
Heat spells = heat waves?
Windthrown is generally caused by wind, so it seems to be a climate risk too.
Reply by authors: Heat spells means “heat waves” or “heat periods” – they are synonyms. Climate risks considered in this study were heath-drought stressors on productivity and tree survival. We have now changed the wording to avoid any confusion (L. 540-542).
Minor comments:
Line 278 – “and fir in Baden-Württemberg. Fig. 1 shows the CAI of A. alba (a) and F. sylvatica (b) for different” – should be Fig 2.
Reply by authors: Corrected
Figure 3 – Fir - no description for RP8 simulation
Reply by authors: Good eye, thank you. Corrected.
Lines 188-189
Table 2: Definition of 10 climate change scenarios with a) base value for atmospheric CO2 concentration (CO2 Base), CO2 increase in % year-1, downregulation factor of photosynthesis, - should be abbreviation, I suppose: PD, increment of temperature (T increase), decrement of precipitation (P decrease) and concentration
Reply by authors: Thank you, corrected
Best regards,
References
Vicente-Serrano, S.M., Beguería, S., López-Moreno, J.I., 2010. A Multiscalar Drought Index Sensitive to Global Warming: The Standardized Precipitation Evapotranspiration Index. J. Clim. 23, 1696–1718. https://doi.org/10.1175/2009JCLI2909.1

Reviewer 2 Report
I commend the authors how they disentangled the effect of climate and CO2-fertilization effects on Fir and Beech species' growth and productivity using GOTILWA+. Forest managers need this kind of information to craft a forest management plan suitable for a future warmer climate. The paper is timely and relevant. However, I have some critical and minor comments to improve this manuscript.
Lines |
Comments |
39-41 |
Intro: The first sentence states a condition when all else (e.g. water availability, etc.) are not limiting. Saying that lines 42-43 contradict the first sentence is a little confusing because the second sentence states a disturbed condition compared to the first sentence where soil water is not limiting. They may only be contradictory if they are of the same condition but different results/impacts on productivity. Please reword. |
175 |
For the benefit of other readers who are not familiar with the term ‘RCP’, kindly spell-out when you first mention it. |
138-139 |
You mentioned you did seasonal field campaigns in spring and summer for photosynthetic and leaf biomass parameters, with 38 samples of each species. Have you measured it during the end of the spring or early spring? What do you think is the cause of no significant differences during leaf initiation and expansion in spring and during summer when leaves are matured? Is having fewer observation data caused no significant differences across seasons in almost all your parameters in lines 238-239? I am a little not convinced of no significant differences across seasons in your study unless it is tropical, perhaps? Hence, the use of constant value along a seasonal development is a little bit maybe unusual for me? Please explain. |
224 |
My modeling knowledge is very basic, but I wonder how was the posterior distribution or frequency distribution of your observed parameters used to constrain in data assimilation. Have they captured well your simulation? What validation method have you applied to validate if your parameters were captured well by your model aside from comparing them from growth and yield tables from other studies? Have you done model forecasting validation? LAI in FigS2 doesn’t seem to constrain well your model (panel b) and may overestimate/underestimate LAI. That is why I asked this because of the use of constant values across seasons. Has this affected your simulation results? |
258-259 |
There is a distinct seasonal pattern in LAI (Fig S2). If LAI varied seasonally, then why not the LMA or the photosynthetic parameters? Is productivity not a function of LAI? I asked this because you said there is no seasonal differences in productivity and other parameters. |
Table 2 |
Be consistent with how many digits you want to use in the table and make it uniform throughout the table. |
278 |
Figure 1 doesn’t show the CAI |
Figure 2 |
Maybe show the medium line in the graph so the reader can imagine if no CC's productivity is slightly above the medium MAI classes. |
280-281 |
“The fluctuations of CAI of the modelled stands are due to fluctuations in the climate file” – How do you confirm this relationship? Is there any regression analysis made? |
Fig S9 |
What is the R2 for the relationship in each scenario |
425 |
‘CO2’ use subscript for 2 |
456-464 |
Maybe it is also worth mentioning the possible effect of evapotranspiration (another component of WUE) on WUE. Evapotranspiration may also be necessary for constraining model simulations. While eCO2 in your study was seen to enhance productivity; however, evapotranspiration may also increase in a future warmer and drier environment. So the trade-off between productivity and transpiration may also be altered. |
Author Response
Response to Reviewer 2
I commend the authors how they disentangled the effect of climate and CO2-fertilization effects on Fir and Beech species' growth and productivity using GOTILWA+. Forest managers need this kind of information to craft a forest management plan suitable for a future warmer climate. The paper is timely and relevant. However, I have some critical and minor comments to improve this manuscript.
Reply by authors: Thank you very much for your positive, constructive feedback and your time dedicated to review our manuscript. We have tried to address all of your comments and hopefully increased clarity. The figure order has been updated and some figures have been improved.
Lines |
Comments |
39-41 |
Intro: The first sentence states a condition when all else (e.g. water availability, etc.) are not limiting. Saying that lines 42-43 contradict the first sentence is a little confusing because the second sentence states a disturbed condition compared to the first sentence where soil water is not limiting. They may only be contradictory if they are of the same condition but different results/impacts on productivity. Please reword. |
Reply by authors |
We agree, that is indeed confusing. We have reworded this sentence now. (L-42-44)
|
175 |
For the benefit of other readers who are not familiar with the term ‘RCP’, kindly spell-out when you first mention it. |
Reply by authors |
Done. (185) |
138-139 |
You mentioned you did seasonal field campaigns in spring and summer for photosynthetic and leaf biomass parameters, with 38 samples of each species. Have you measured it during the end of the spring or early spring? What do you think is the cause of no significant differences during leaf initiation and expansion in spring and during summer when leaves are matured? Is having fewer observation data caused no significant differences across seasons in almost all your parameters in lines 238-239? I am a little not convinced of no significant differences across seasons in your study unless it is tropical, perhaps? Hence, the use of constant value along a seasonal development is a little bit maybe unusual for me? Please explain. |
Reply by authors |
Our field campaigns started beginning of June and beginning of august (08.06.–16.06.2017 and 10.08.-18.08.2017). We needed maximum, potential parameters for the model. If environmental conditions (drought, heat, nutrients) do not impose any limitation, I do not expect the photosynthetic potentials or leaf morphology to vary much between peak of spring to peak of summer. Additionally, the measurement year 2017 was a very wet year. See also Fig. 4 in (Wilson et al., 2001). We were probably still in the peak phase and even though the values started declining, the change was not yet significant. Beginning of spring until full development there are certainly changes, just as at the beginning of leaf senescence late summer/beginning of autumn. We however, needed the maximum at the peak of the vegetation for the parametrization of GOTILWA+ because the rest is simulated by the phenology sub-module. Regarding the question is whether these parameters vary seasonally once leaves are unfolded or if they stay constant – as you said for tropical forests: In previous studies, I have investigated in-depth the seasonality in Mediterranean, semi-arid ecosystems but also temperature forests. I found a great variability between the different seasons in Mediterranean conditions, mainly due to changes in water supply combined with heat stress (Sperlich et al., 2015). In temperate ecosystems, changes in water supply and heat stress were less a critical issue (expect for particular drought years) and the photosynthesis and leaf biomass parameters are more constant during the growth period (see for example). Additionally, the year of measurement 2017 was quite a wet year – especially in summer – so to me it is not surprising that we did not see seasonal changes because spring and summer were not much different. In the tropics, it is similar: there can be seasonal differences in leaf morphological and photosynthesis parameters if there is a pronounced rainy season and a drier season (Brodribb et al., 2002), and also trees may shedd their leaves in order to prevent catastrophic xylematic damage under drought stress(Wolfe et al., 2016). Otherwise, it depends more on leaf ageing than environmental variables. See also discussion in (Sperlich et al., 2019, 2015). Our sentence regarding constant values of photosynthetic potentials refer to the fact that Vc,max and Jmax are thus often used as constants for various plant functional types and seasons or, in some cases, are derived from other parameters such as leaf nitrogen content (Grassi and Magnani, 2005; Walker et al., 2014) (see also discussion “Implications for the global carbon cycle and modelling” in Sperlich et al., 2015). In GOTILWA+, however, water stress directly reduces the photosynthetic potential through a nonlinear relation to soil water content by using an empirical β coefficient (suppl. mat. Note S4), which is different to many models which have not implemented such a mechanistic link of drought with the photosynthesis sub-module (Drake et al., 2017; Eller et al., 2020). Long story short, against this background our sentence is quite confusing and we have deleted it now.
|
224 |
My modeling knowledge is very basic, but I wonder how was the posterior distribution or frequency distribution of your observed parameters used to constrain in data assimilation. Have they captured well your simulation? What validation method have you applied to validate if your parameters were captured well by your model aside from comparing them from growth and yield tables from other studies? Have you done model forecasting validation? LAI in FigS2 doesn’t seem to constrain well your model (panel b) and may overestimate/underestimate LAI. That is why I asked this because of the use of constant values across seasons. Has this affected your simulation results? |
Reply by authors |
We have mainly used the productivity of the long-term experimental plots. Additionally, we have conducted extensive inventories along different age-class distribution to constrain the model regarding tree density and standing volume (Fig. S3, suppl. mat.). Regarding LAI: LAI development is quite different for deciduous and evergreen tree species. While the model captures well the seasonal evolution of the former, it does not for the latter. The reason is twofold. I was rather surprised to see such a strong seasonal change in measured LAI because I expected it to be more homogenous for evergreen species such as the conifer Abies alba. It may also be an artefact of the indirect measurement technique with the optical sensor of LAI-2200C, which is generally problematic for conifers. However, that opens a new discussion that we cannot address in this paper (see e.g. Bréda, 2003). The other reason is the representation of coniferous species in the model. Leaf ageing and development of new leaves depends on leaf age (in this study assumed 5 years for Abies Alba). While old leaves are shed, new leaves replace the old leaves, but this does not become visible in the simulated LAI and Fig S2 panel b. The LAI of conifers is thus difficult to represent in the model also because it is hard to get the true LAI of conifers. In the end, we had to calibrate the LAI using a mean value (not using the peak in summer and not using the lower values before and after). Our aim was to represent a typical productivity class of Abies alba for the study region, and we managed that quite well. “The photosynthetic submodule together with the leaf biomass parameters leaf mass per leaf area (LMA, g cm-2) and leaf area index (LAI in m2 m-2) are critical parameters parameters for the efficiency of total canopy carbon gain and, together with respiration, for forest growth and productivity (supplementary material Note S4.).” Saying this, we eventually fitted the LAI of fir and calibrated it together with photosynthesis and LMA using the upper and lower bounds of the measured values. GOTILWA+ regulates the photosynthetic carbon gain of conifers with the photosynthetic submodule and not with seasonal leaf fall represented by declining LAI. We validated the result with current annual increment over stand age – our productivity indicator – in Fig. 2 (now Fig.3). Also, we used the stem density of different inventory plots in the study region to create realistic tree numbers and thinnings so that we can be sure our simulated stands represents a realistic, medium productivity stand in the study region. We have complemented this validation with a new panel (a2 and b2) in Fig. 3, showing the CAI of the simulated stands overs the CAI of the experimental plots by FVA. |
258-259 |
There is a distinct seasonal pattern in LAI (Fig S2). If LAI varied seasonally, then why not the LMA or the photosynthetic parameters? Is productivity not a function of LAI? I asked this because you said there is no seasonal differences in productivity and other parameters. |
Reply by authors |
We have addressed this in the above question. Concluding the above, GOTILWA+ does not regulate the seasonality in carbon gain of evergreen conifers with the seasonality in LAI, but with the seasonality in the photosynthesis sub-module. We have now added a paragraph to explain this better (L.277-288). |
Table 2 |
Be consistent with how many digits you want to use in the table and make it uniform throughout the table. |
Reply by authors |
Corrected. |
278 |
Figure 1 doesn’t show the CAI |
Reply by authors |
Corrected. We have updated the figure order now, as there were some more changes. |
Figure 2 |
Maybe show the medium line in the graph so the reader can imagine if no CC's productivity is slightly above the medium MAI classes. |
Reply by authors |
We believe it is better not to highlight the medium productivity line because we are looking at a single stand while the yield table represents information from many long-term experimental plots from FVA. There is quite some variability in our simulated CAI due to the climate data used for our plot, while there is not for the yield table data of FVA because those are averaged values. However, the grey lines in the background from the yield tables indicate nicely that our simulated stand is in the range of what is realistic in the study region. However, we have added panel a2 and b2 in Fig. 3 to showt the relationship of simulation with the medium/good productivity class. |
280-281 |
“The fluctuations of CAI of the modelled stands are due to fluctuations in the climate file” – How do you confirm this relationship? Is there any regression analysis made? |
Reply by authors |
There is a strong relationship of productivity with drought. We analysed productivity with NPP in relation to soil water content (SWC) in Fig. S9 (supp. mat). Fluctuations of SWC are a result of climate fluctuations. We have also looked at the marker years when CAI fall to a minimum value (stand age 105 and 110, simulation year 2105 and 2110) and found that precipitation was particularly low in simulation year 2103-2105. We have reformulated this sentence now and pointing to fig. S9. We also gave additional information regarding the marker year. |
Fig S9 |
What is the R2 for the relationship in each scenario |
Reply by authors |
We have now added the R2 and the equation the figure caption of Fig. S9 |
425 |
‘CO2’ use subscript for 2 |
Reply by authors |
Corrected. |
456-464 |
Maybe it is also worth mentioning the possible effect of evapotranspiration (another component of WUE) on WUE. Evapotranspiration may also be necessary for constraining model simulations. While eCO2 in your study was seen to enhance productivity; however, evapotranspiration may also increase in a future warmer and drier environment. So the trade-off between productivity and transpiration may also be altered. |
Reply by authors |
Thank you for this valuable remark. We have now mentioned the potential trade-off between productivity and evapotranspiration. (L. 485-487) |
References
Bréda, N.J.J., 2003. Ground-based measurements of leaf area index: a review of methods, instruments and current controversies. J. Exp. Bot. 54, 2403–17. https://doi.org/10.1093/jxb/erg263
Brodribb, T.J., Holbrook, N.M., Gutiérrez, M. V., 2002. Hydraulic and photosynthetic co-ordination in seasonally dry tropical forest trees. Plant, Cell Environ. 25, 1435–1444. https://doi.org/10.1046/j.1365-3040.2002.00919.x
Drake, J.E., Power, S.A., Duursma, R.A., Medlyn, B.E., Aspinwall, M.J., Choat, B., Creek, D., Eamus, D., Maier, C., Pfautsch, S., Smith, R.A., Tjoelker, M.G., Tissue, D.T., 2017. Stomatal and non-stomatal limitations of photosynthesis for four tree species under drought: A comparison of model formulations. Agric. For. Meteorol. 247, 454–466. https://doi.org/10.1016/j.agrformet.2017.08.026
Eller, C.B., Rowland, L., Mencuccini, M., Rosas, T., Williams, K., Harper, A., Medlyn, B.E., Wagner, Y., Klein, T., Teodoro, G.S., Oliveira, R.S., Matos, I.S., Rosado, B.H.P., Fuchs, K., Wohlfahrt, G., Montagnani, L., Meir, P., Sitch, S., Cox, P.M., 2020. Stomatal optimization based on xylem hydraulics (SOX) improves land surface model simulation of vegetation responses to climate. New Phytol. 226, 1622–1637. https://doi.org/10.1111/nph.16419
Grassi, G., Magnani, F., 2005. Stomatal, mesophyll conductance and biochemical limitations to photosynthesis as affected by drought and leaf ontogeny in ash and oak trees. Plant, Cell Environ. 28, 834–849. https://doi.org/10.1111/j.1365-3040.2005.01333.x
Sperlich, D., Chang, C.T., Peñuelas, J., Gracia, C., Sabaté, S., 2015. Seasonal variability of foliar photosynthetic and morphological traits and drought impacts in a Mediterranean mixed forest. Tree Physiol. 35, 501–520. https://doi.org/10.1093/treephys/tpv017
Sperlich, D., Chang, C.T., Peñuelas, J., Sabaté, S., 2019. Responses of photosynthesis and component processes to drought and temperature stress: Are Mediterranean trees fit for climate change? Tree Physiol. 39, 1783–1805. https://doi.org/10.1093/treephys/tpz089
Walker, A.P., Beckerman, A.P., Gu, L., Kattge, J., Cernusak, L. a., Domingues, T.F., Scales, J.C., Wohlfahrt, G., Wullschleger, S.D., Woodward, F.I., 2014. The relationship of leaf photosynthetic traits - Vcmax and Jmax - to leaf nitrogen, leaf phosphorus, and specific leaf area: a meta-analysis and modeling study. Ecol. Evol. 4, 3218–3235. https://doi.org/10.1002/ece3.1173
Wilson, K.B., Baldocchi, D.D., Hanson, P.J., Ridge, O., 2001. Leaf age affects the seasonal pattern of photosynthetic capacity and net ecosystem exchange of carbon in a deciduous forest. Plant, Cell Environ. 24, 571–583.
Wolfe, B.T., Sperry, J.S., Kursar, T.A., 2016. Does leaf shedding protect stems from cavitation during seasonal droughts? A test of the hydraulic fuse hypothesis. New Phytol. 212, 1007–1018. https://doi.org/10.1111/nph.14087

Reviewer 3 Report
Dear Authors,
in the ms you combine field data and different climatic chane scenarios
Your ms is very interesting especially for forest managers to make decisions. Althought your ms does not result in a clear outcome for the use of the two species your study gives an isight into the impact of climate change on forest productivity and mortality in relation to climatic change. Probably with some modifications your climatic change scenarios that you have developed could be used also in other natural ecosystem to study their vulnerability to climatic change.
A shortcoming in your ms: In figures 5 and S5 and especially in fig a3 and b3 there is difference in horizontal axis scale (stand age)
Author Response
Reviewer 1:
Dear Authors,
in the ms you combine field data and different climatic chane scenarios
Your ms is very interesting especially for forest managers to make decisions. Althought your ms does not result in a clear outcome for the use of the two species your study gives an isight into the impact of climate change on forest productivity and mortality in relation to climatic change. Probably with some modifications your climatic change scenarios that you have developed could be used also in other natural ecosystem to study their vulnerability to climatic change.
A shortcoming in your ms: In figures 5 and S5 and especially in fig a3 and b3 there is difference in horizontal axis scale (stand age)
Reply by authors: Thank you very much for your positive, constructive feedback and your time dedicated to review our manuscript. We have corrected and improved the figures and also updated the figure order now.
Round 2
Reviewer 1 Report
Overall, I found that the manuscript has been improved. My comments and suggestions have been included in the revised version. However, I found a few editorial errors:
lines 263-264 – there is table 2, should be Table 3,
Please, check the references in the text to Figures 4-7.
Best regards,
Author Response
We thank reviewer 1 for the positive feedback. We have fixed now the editorial errors.
Best,
Reviewer 2 Report
Thank you for excellently addressing all my concerns.
Author Response
Thank you very much for your positive feedback.